# Anatomy-guided Latent Diffusion Model for Fine-grained Medical Image Synthetic Augmentation

**Sang-Heon Lim**[*1]                                    SMION123@SNU.AC.KR
[1]*Interdisciplinary Program of Bioengineering, Seoul National University, Seoul, South Korea*
**Su Yang**[*2]                                          S8431@SNU.AC.KR
[2]*Department of Applied Bioengineering, Graduate School of Convergence Science and Technology, Seoul National University, Seoul, South Korea*
**Jiyong Han**[1]                                        JIYONG@SNU.AC.KR
**SuJeong Kim**[1]                                       SUJEONG@SNU.AC.KR
**Da-El Kim**[1]                                         DKIM3@SNU.AC.KR
**Yu-Ri Kim**[3]                                         9651230@NAVER.COM
[3]*Department of Oral and Maxillofacial Radiology, School of Dentistry and Dental Research Institute, Seoul National University, Seoul, South Korea*
**Jun-Min Kim**[4]                                       JMKIM@HANSUNG.AC.KR
[4]*Department of Electronics and Information Engineering, Hansung University, Seoul, Korea*
**Jo-Eun Kim**[3]                                        NOEL1ST@SNU.AC.KR
**Won-Jin Yi**[†1,2,3]                                   WJYI@SNU.AC.KR

**Editors:** Under Review for MIDL 2024

## Abstract

Medical data typically requires expert annotation to produce a reliable quantitative organ analysis, which can be costly and time-consuming. Recently, several deep learning-based synthetic augmentations have been proposed to address the limitations. However, previous success of generative synthetic augmentation methods cannot be guaranteed without additional fine-tuning. To mitigate the dependency on this issue, we propose an anatomy-guided latent diffusion model, which can perform anatomical synthesis in a selectively latent blending manner. We evaluate the proposed approach using a mandibular canal segmentation dataset on panoramic dental radiographs. The segmentation performance was improved by a Dice similarity coefficient of 16.6% with our proposed synthetic augmentation.

**Keywords:** latent diffusion model, latent blending, anatomical synthesis

## 1. Introduction

One of the primary challenges for deep learning in the medical domain is the scarcity of accessible datasets with expert annotation. Constructing reliable labeled datasets is time-consuming, labor-intensive, and demands significant domain knowledge. Recently, diffusion model (DM)-based synthetic augmentation studies (Ye et al., 2023; Oh and Jeong, 2023) have been proposed to overcome these challenges. However, the previous studies depend on further fine-tuning for image generation. Furthermore, unlike latent diffusion models (LDMs), basic DMs require significant computational resources because they operate in the full data space.

---

[*] Contributed equally
[†] Corresponding author

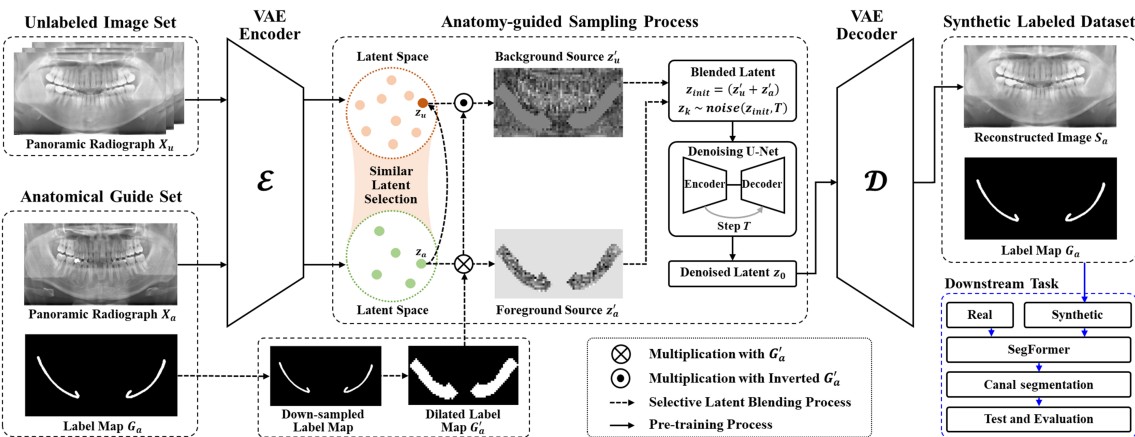

Figure 1: Framework of anatomy-guided latent bleding-based LDM for synthetic data augmentation.

In this study, therefore, we propose an unconditional LDM-based anatomy-guided latent blending strategy to generate fine-grained anatomical synthetic data. Instead of optimizing a mask-conditional LDM, we leverage an unconditional diffusion probabilistic model (DPM) for the reverse diffusion iterations applied to the given blended latent representations. This study aims to generate a synthetic dataset for mandibular canal segmentation that enables few-shot anatomical segmentation on panoramic radiographs (PANs).

## 2. Methods

**Data configuration.** This study was approved by the Institutional Review Board of Seoul National University Dental Hospital (No. ERI23015). A total of 7,263 PANs were acquired, comprising labeled 2,100 PANs (called the *anatomical guide set*) and an unlabeled 5,163 PANs. We used an unlabeled data set consisting of 3,613 images for the training of the LDM and 1,550 images for the synthetic augmentation. In addition, 210 images from the labeled dataset were used for synthetic augmentation, while the 1,890 images constituted a hold-out test set for evaluation. All data were resized to $512 \times 256$ and normalized to [-1, 1]. We also used a pre-trained buccal segmentation model to remove unnecessary regions.

**Anatomy-guided latent blending.** We used anatomy-guided LDM to generate synthetic PANs that could be utilized to image-mask paired datasets without further manual segmentation (Figure 1). To prevent data leakage, the data used for synthesis was excluded from the LDM training set. First, the pre-trained variational autoencoder (VAE) encoded unlabeled PANs $X_u$ and labeled PANs $X_a$ as lower-dimensional latent representations to obtain each latent space. In the latent spaces, the structural similarity index measure (SSIM) was utilized for the selection of similar latents of $z_u$ and $z_a$, leading to anatomical latent blending. We hypothesized that this approach for latent selection would prevent the blending of unrelated latents and improve the stability of the sampling process. As denoted in Appendix A, the $G_a$ derived from the *anatomical guide set* was used to synthesize the anatomical structure within the unlabeled PANs selectively.

Following the previous latent blending study (Avrahami et al., 2023), we down-sampled the binary label map $G_a$ of size $64 \times 32$ for use as anatomical guidance for each latent $z_u$ and

Table 1: Quantitative comparison results of synthetic augmented segmentation. "Real" refers to a real dataset while "synthetic" indicates an anatomy-guided synthetic dataset.

| Training data composition | | | Evaluation metrics | | |
|---|---|---|---|---|---|
| Latent selection | Real | Synthetic | Precision | Recall | DSC |
| - | ✓ | | 0.593±0.172 | 0.411±0.174 | 0.477±0.172 |
| SSIM Top 5 | | ✓ | 0.591±0.173 | 0.324±0.143 | 0.411±0.156 |
| ($n = 1,050$) | ✓ | ✓ | 0.661±0.153 | 0.522±0.157 | 0.579±0.152 |
| SSIM Top 10 | | ✓ | 0.625±0.154 | 0.501±0.150 | 0.552±0.148 |
| ($n = 2,100$) | ✓ | ✓ | 0.684±0.149 | 0.594±0.148 | 0.633±0.144 |
| SSIM Top 20 | | ✓ | 0.663±0.164 | 0.573.±0.157 | 0.611±0.154 |
| ($n = 4,200$) | ✓ | ✓ | **0.689±0.156** | **0.608±0.152** | **0.643±0.149** |
| SSIM Top 50 | | ✓ | 0.639±0.169 | 0.577±0.164 | 0.603±0.162 |
| ($n = 10,500$) | ✓ | ✓ | 0.659±0.162 | 0.612±0.152 | 0.632±0.152 |

$z_a$. Furthermore, we obtained dilated label map $G'_a$ by performing dilation filtering on $G_a$ to mitigate information imbalance of the narrow mandibular canal regions in background source $z'_u$ and foreground source $z'_a$, where the $z'_u$ and $z'_a$ were obtained using element-wise multiplication with $G'_a$ and inverted $G'_a$. We used the blended latent of $z'_u$ and $z'_a$ as the initial latent $z_{init}$. The denoised latent $z_0$ was obtained with 25 times of reverse diffusion steps. The $z_0$ is then passed into the VAE decoder to reconstruct the synthetic image $S_a$. To assess the reliability of our synthetic augmentation approach, we exploited the SegFormer (Xie et al., 2021) for the downstream mandibular canal segmentation task.

**Implementation details.** We designed our model based on Medfusion (Müller-Franzes et al., 2023). We used the AdamW optimizer for network training and linear noise scheduling for the forward diffusion process. The total number of forward diffusion steps was 1,000, and VAE was trained for 2,000 epochs and an unconditional denoising U-Net for 1,000 epochs using an NVIDIA RTX A6000 GPU.

## 3. Results and Discussion

In this study, the top 5, 10, 20, and 50 unlabeled PANs most similar to the image from *anatomical guide set* were selected for synthetic image sampling. Applying anatomy-guided synthesis with selected unlabeled PANs, we obtained four different synthetic datasets comprising paired images and masks of 1,050, 2,100, 4,200, and 10,500. The examples of anatomical synthetic data were presented in Appendix B.

Table 1 shows the performance comparison of mandibular canal segmentation using real datasets and synthetic augmented datasets. We observed that anatomy-guided synthetic augmentation improves the performance of canal segmentation in all situations. Additionally, we conducted a further evaluation with the model trained with a synthetic dataset alone. Building on this evaluation, we demonstrated that our approach can perform fine-grained anatomical synthesis from unlabeled data and make it usable as a reliable augmented training dataset. In conclusion, we successfully generated a synthetic dataset for mandibular canal segmentation on PANs based on anatomy-guided latent blending LDM. We believe that our model may provide a fine-grained synthesis of the desired region of interest with low distortion of the original unlabeled image.

## Acknowledgments

This study was supported by a Korea Medical Device Development Fund Grant by the Korean government (Ministry of Science and ICT; Ministry of Trade, Industry, and Energy; Ministry of Health and Welfare; Ministry of Food and Drug Safety) (Project Number:1711194231, KMDF_PR_20200901_0011).

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

## Appendix A. Sampling process with anatomy-guided latent blending.

**Require:** Trained VAE encoder $\mathcal{E}(x)$ and VAE decoder $\mathcal{D}(z)$.
**Input:** unlabeled image $X_u$, labeled image $X_a$, binary label map $G_a$, denoising step $T$.
**Output:** synthetic image $S_a$.

$\quad z_u, z_a = \mathcal{E}(X_u), \mathcal{E}(X_a)$
$\quad G'_a = Downsample(G_a) * DilationFilter$
$\quad z'_u, z'_a = z_u \times G'_a, z_a \times (1 - G'_a)$
$\quad z_{init} \sim z'_u + z'_a$
$\quad z_k \sim noise(z_{init}, T)$
$\quad \textbf{for } t = T, ..., 1 \textbf{ do}$
$\quad\quad\quad z_{t-1} \leftarrow denoise(z_t)$
$\quad \textbf{end}$
$\quad S_a = \mathcal{D}(z_0)$
$\quad \textbf{return } S_a$

## Appendix B. Representative examples of anatomy-guided synthetic data.

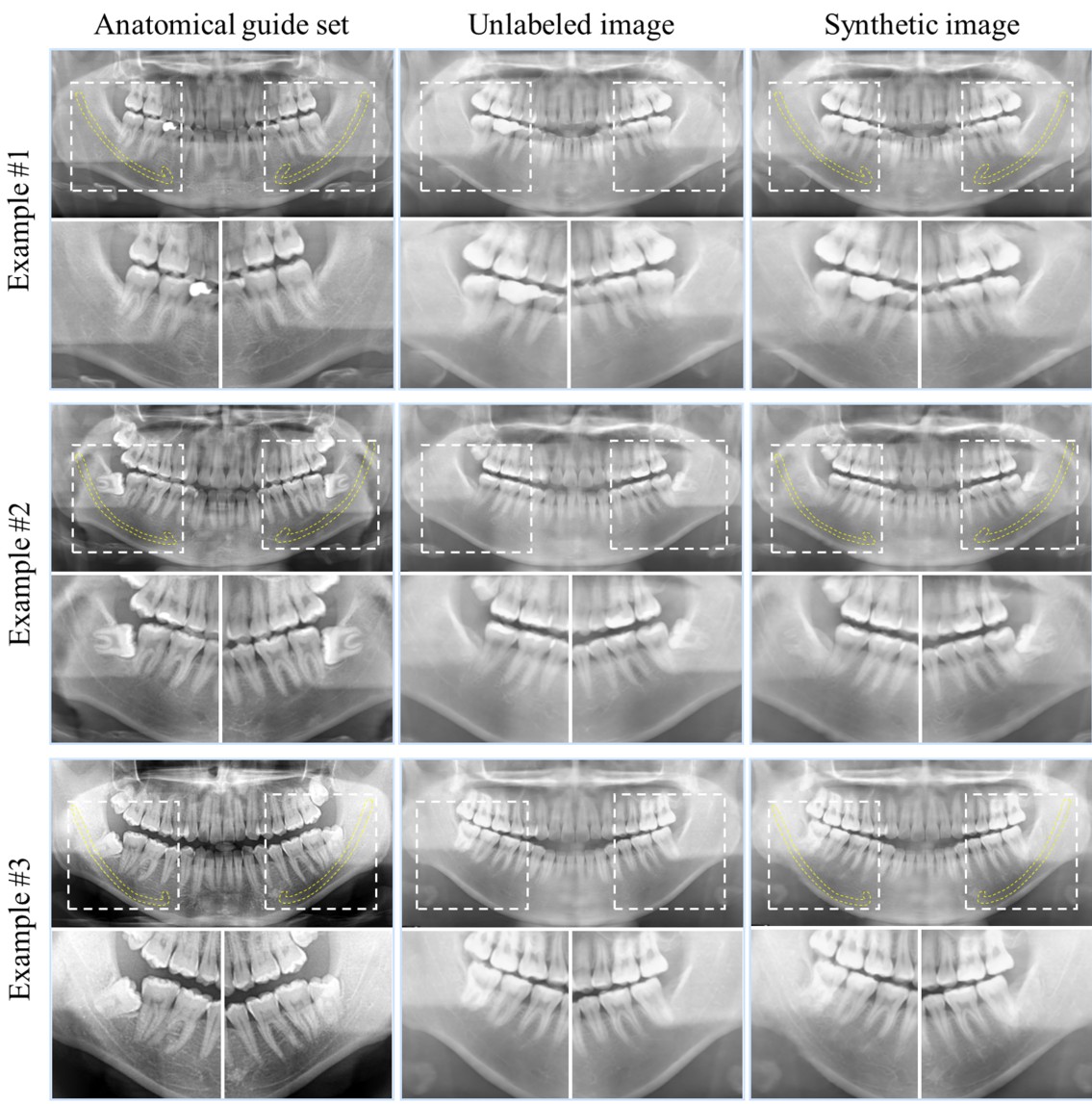

