# OpenReview forum: "Anatomy-guided Latent Diffusion Model for Fine-grained Medical Image Synthetic Augmentation"
_MIDL.io/2024/Short_Papers — MIDL 2024 Short Papers_

### Official Review · Reviewer_d2Bs · 2024-04-24

**Confidence:** 4
**Final Rating:** 5

**Review:**

This paper introduces a new approach utilizing a latent diffusion model for synthesizing image data with the guidance of anatomical structures (represented by region-of-interest segmentation label maps). The authors evaluate the effectiveness of the proposed method on a mandibular canal segmentation dataset from panoramic dental radiographs. Experimental results showcase an improved performance through dice metrics. Despite the constraints of space in MIDL short papers, the paper is well-written, and the experimental evaluation provides convincing evidence of the method's efficiency.

---

### Decision · Program_Chairs · 2024-04-26

Accept